# Therapeutic Potential of a Natural Blend of *Aronia melancarpa*, *Lonicera caerulea*, and *Echinacea purpurea* Extracts in Treating Upper Respiratory Tract Infections: Preliminary Clinical and In Vitro Immunomodulatory Insights

**DOI:** 10.3390/ijms252413436

**Published:** 2024-12-15

**Authors:** Katarzyna Zima, Marta Sochocka, Michał Ochnik, Barbara Khaidakov, Krzysztof Lemke, Paulina Kowalczyk

**Affiliations:** 1Department of Physiology, Medical University of Gdańsk, Dębinki Street 1, 80-211 Gdańsk, Poland; katarzyna.zima@gumed.edu.pl; 2Laboratory of Virology, Hirszfeld Institute of Immunology and Experimental Therapy, Polish Academy of Sciences, Weigla Street 12, 53-114 Wrocław, Poland; marta.sochocka@hirszfeld.pl (M.S.); michal.ochnik@hirszfeld.pl (M.O.); 3R&D Department, AronPharma Ltd., Trzy Lipy Street 3, 80-172 Gdańsk, Poland; barbara.khaidakov@aronpharma.pl (B.K.); krzysztof.lemke@aronpharma.pl (K.L.); 43P-Medicine Laboratory, Medical University of Gdańsk, Dębinki Street 7, 80-211 Gdańsk, Poland

**Keywords:** immunomodulatory, antiviral, polyphenols, Rosaceae, Caprifoliaceae, Asteraceae

## Abstract

Upper respiratory tract infections (URTIs) are a prevalent health issue, causing considerable morbidity. Despite the availability of conventional treatments, there is an increasing interest in natural products due to their potential antiviral and immunomodulatory benefits. This study aims to evaluate the efficacy of an ELA blend (E—*Echinacea purpurea*, L—*Lonicera cerulea*, A—*Aronia melanocarpa*) in preventing and alleviating the symptoms of URTIs. Additionally, the study examines the blend’s antiviral and immunomodulatory effects both in vitro and through a clinical trial. A randomized, double-blind, placebo-controlled trial involved 61 participants prone to URTIs, with a 60-day treatment and follow-up period. A placebo group later received the ELA blend for 60 days. The ELA blend significantly reduced the incidence of URTIs during the observation period (2 vs. 8; *p* = 0.044) and, in particular, throat-related symptoms (8 vs. 16; *p* = 0.038). Analyses of PBMCs showed that baseline production of the cytokines IFN-γ (*p* = 0.020), IL-1β (*p* = 0.004), IL-2(*p* < 0.001), IL-6 (*p* < 0.001), and TNF-α (*p* < 0.001) increased after ELA blend treatment. Moreover, the ELA blend modulated cytokine production in response to PHA-L stimulation, decreasing IFN-γ (*p* = 0.008) and IL-2 (*p* = 0.012) while increasing IL-1β (*p* = 0.005). Following R848 stimulation, the ELA blend enhanced the production of INF-α (*p* = 0.012) and IL-2 (*p* = 0.025), and decreased IL-1β (*p* < 0.001), IL-6 (*p* < 0.001), and TNF-α (*p* = 0.049). The blend suppressed VSV replication and significantly increased cytokine levels, with IFN-γ increasing by 98 pg/mL (*p* = 0.002), IL-1β rising by 233.0 pg/mL (*p* = 0.004), and TNF-α showing an increase of 2905 pg/mL (*p* = 0.002). These findings highlight the ELA blend’s potential to alleviate URTI symptoms, modulate inflammatory and antiviral immune responses, and inhibit viral replication. Further investigations should aim to validate these findings through large-scale studies, and explore the ELA blend’s long-term safety and efficacy in diverse populations. Additionally, research should investigate optimal dosing strategies and explore potential synergistic effects with conventional treatments to maximize clinical outcomes. Trial registration: retrospectively registered under NCT06020001.

## 1. Introduction

Infections of the upper respiratory tract, including those caused by SARS-CoV-2, present a substantial global health challenge, characterized by symptoms such as a cough, a sore throat, nasal congestion, and respiratory distress [1]. The ongoing COVID-19 pandemic underscores the urgent need for novel therapeutic strategies that leverage natural compounds to prevent and treat upper respiratory tract infections (URTIs) [2,3]. Plants have long been considered as potential sources of molecules which may be useful in the management or prevention of many human pathologies with inflammatory components, such as viral infections [4,5]. Single natural-origin compounds are characterized by diversity in their multiple known mechanisms of action, but often have limited possibilities of use. Promising options may include natural extract blends. However, much remains to be understood about their mechanisms of action and influence on human health. Medicinal plant-derived blends may demonstrate lower toxicity, stronger beneficial antioxidants, and stronger immunomodulatory or antiviral activity than synthetic pharmaceuticals [6].

*Aronia melanocarpa* (chokeberry), *Lonicera caerulea* (Kamchatka honeysuckle), and *Echinacea purpurea* (purple coneflower) are three polyphenol-rich plants that have garnered international attention for their health-promoting properties. *Aronia melanocarpa* is a well-studied species which exhibits potent antioxidant and anti-inflammatory properties. In vitro studies have shown that aronia extracts can inhibit the production of nitric oxide (NO) and pro-inflammatory cytokines, such as IL-10 and (tumor necrosis factor α) TNF-α, in human monocytic MM6 cells and RAW 264.7 macrophages stimulated with lipopolysaccharide (LPS) [7,8]. Further in vitro studies have demonstrated that aronia juice can inhibit the replication of various influenza virus strains, including those resistant to antiviral drugs such as oseltamivir [9]. Additionally, aronia extracts have been shown to modulate immune responses in vivo, reducing inflammation in experimental models of eye inflammation, and suppressing the activation of nuclear factor kappa-light-chain-enhancer of activated B cells (NF-κB), a key inflammatory pathway [10,11]. This indicates that aronia could serve as a beneficial dietary supplement or therapeutic agent, especially in contexts where inflammation and viral infections are prevalent.

*Lonicera caerulea* (blue honeysuckle) has also gained attention for its immunomodulatory properties. Research indicates that its phenolic extracts exhibit antibacterial activity, which may be valuable in controlling infections, particularly respiratory pathogens [12]. The extracts of Lonicera caerulea berries have shown potential in reducing inflammatory markers, such as Monocyte Chemoattractant Protein-1 (MCP-1), IL-10, and various interleukins (IL-6, IL-4, IL-12, TNF-α), in murine models [13], suggesting that this plant could play a role in modulating immune responses and supporting overall health.

*Echinacea purpurea*, commonly known as purple coneflower, is renowned for its traditional use in treating URTIs. Clinical studies have demonstrated its efficacy in preventing and managing URTIs, partly due to its ability to enhance immune cell activity, such as macrophage and T-cell function [14,15]. Moreover, Echinacea’s active constituents, such as alkamides, stimulate macrophage activity and increase phagocytosis, which are crucial for combating bacterial and viral infections [16].

Together, these plants offer a promising alternative or complementary approach to managing immune-related disorders, particularly during viral outbreaks like COVID-19, when supporting the immune system is critical. Our earlier research demonstrated that a polyphenol-rich ELA blend of LCK, AM, and EP exhibited potent effects on key cellular mechanisms involved in viral infections [17].

We hypothesize that the ELA blend, comprising extracts from chokeberry (*Aronia melanocarpa*), Kamchatka honeysuckle (*Lonicera caerulea* var. *kamtschatica*), and purple coneflower (*Echinacea purpurea*), demonstrates efficacy in reducing the incidence, severity, and duration of URTIs in individuals susceptible to such infections in a clinical setting. Additionally, our study explores the blend’s antiviral and immunomodulatory properties through in vitro and ex vivo analyses of peripheral blood immune cells, providing complementary mechanistic insights.

## 2. Results

### 2.1. Study Population

The study population consisted of 61 patients randomly assigned to two groups: 30 patients received the placebo/syrup, and 31 patients received the capsule formulation (Table 1). The distribution of sex between the groups was not statistically different. The median age of all patients was 59 years in both the placebo/syrup group and the capsule group. When stratified by sex, the women in the placebo/syrup group were younger than the women in the capsule group. The difference in age between the female groups was statistically significant (*p* = 0.005). The men in the placebo/syrup group were older than the men in the capsule group, with no significant difference in age observed between the male groups (*p* = 0.355). Patients in both the placebo/syrup and capsule groups reported a comparable number of colds or URTIs during the three months preceding the study, with no significant difference between the groups (*p* = 0.585). Similarly, the prevalence of comorbidities was evenly distributed between the groups, with conditions such as hypertension, diabetes, and allergic diseases occurring at similar rates, and no statistically significant differences were observed in comorbidity prevalence (all *p* > 0.999).

### 2.2. Influence of the ELA Blend in Capsule and Syrup Form on the Frequency and Duration of Symptoms

In this study, the outcomes of capsule treatment (ELA blend, 300 mg, twice a day) were assessed in comparison to a placebo in a cohort of patients. The analysis revealed notable differences in symptom reporting between the two groups (Table 2). Specifically, during the observation period (D1–D120), a significantly lower proportion of patients in the blend group reported experiencing at least three out of four concurrent symptoms (runny nose, cough, fever, difficulty breathing) compared to the placebo group (*p* = 0.044). Similarly, during the treatment phase (D1–D60), fewer patients in the ELA blend group reported a sore throat, hoarseness, or throat scratching compared to the placebo group (*p* = 0.038). Furthermore, the incidence of fever occurrence during supplementation (number of patients reporting symptoms multiplied by the number of days with symptoms) was notably lower in the capsule group compared to the placebo group (2 vs. 9; *p* = 0.039). Additionally, the occurrences of sore throat/hoarseness/throat scratching during supplementation were substantially lower in the capsule group compared to the placebo group (32 vs. 85; *p* < 0.001). Appendix A illustrates the number of symptom reports on the day of reporting and the intensity of these symptoms, classified by group (findings from the complete study D1–D120).

In the second phase of the study, the syrup group and placebo group represent the same individuals who underwent both phases of the trial sequentially. Initially, these patients were assigned to the placebo group and received placebo treatment. Subsequently, after a designated period, the same individuals were then allocated to the syrup group and received the experimental syrup formulation containing the same dosage of the blend (300 mg twice a day) as the capsules. This sequential design allows for within-patient comparisons, enabling a direct assessment of the effects of the syrup treatment relative to the placebo within the same cohort of patients. During the syrup treatment phase, it was observed that the same patients who had previously received the placebo reported fewer instances of experiencing at least three out of four symptoms simultaneously (runny nose, cough, fever, difficulty breathing) compared to their earlier placebo phase. Importantly, no patients in the syrup group reported fever during treatment, whereas four patients had reported fever during the preceding placebo phase. Additionally, fewer patients from the same cohort reported sore throat, hoarseness, or throat scratching, and cough and runny nose, stuffy nose, or sneezing during syrup treatment compared to their previous placebo phase. Appendix A illustrates the number of symptom reports on the day of reporting and the intensity of these symptoms (findings from the complete study D1–D180).

These results highlight the potential benefits of ELA blend supplementation in reducing the frequency, severity, and duration of symptoms during and after the treatment period, emphasizing its clinical relevance in managing respiratory symptoms.

### 2.3. Influence of ELA Blend on Serum Parameters and Cytokine Production in Stimulated Peripheral Blood Mononuclear Cells (PBMCs) Isolated from Study Participants

To investigate the immunological effects of the ELA blend, we assessed serum CRP concentration and SOD activity, and analyzed cytokine production, in PBMCs stimulated with PHA-L (Phytohemagglutinin-L) or R848 (Resiquimod) (Table 3). Initially, there were no significant differences in these parameters between the capsule and placebo groups at the beginning of the study (D0). During the D0 to D60 period, we observed that serum CRP levels did not significantly differ between the capsule and placebo groups at D60 (*p* = 0.17). Both groups showed a median increase in C-Reactive Protein (CRP) from D0 to D60. Similarly, Superoxide Dismutase (SOD) levels in serum did not significantly differ between the two groups at D60 (*p* = 0.303).

Further analysis of Interferon-alpha (IFN-α) production in PBMCs stimulated with R848 did not reveal a significant difference between the capsule and placebo groups after treatment (*p* = 0.407). Regarding cytokine production by PBMCs stimulated with PHA-L at D60, specifically for IL-6 and TNF-α, no significant differences were observed between the groups at D60, or within the groups before and after treatment. However, the levels of IL-1β were higher in the capsule group compared to the placebo group. Both groups showed an increase in IL-1β production by PHA-L-stimulated PBMCs from D0 to D60, with a larger median difference observed in the capsule group compared to the placebo group. Overall, these findings suggest that, aside from the observed increase in IL-1β production in the capsule group, the capsules did not significantly alter serum SOD and CRP levels or cytokine production compared to placebo during the study period.

Analysis of parameters in the syrup group before and after supplementation (D120 vs. D180) revealed several outcomes (Table 4). There were no significant differences in CRP levels and SOD activity in serum between D120 and D180 (CRP: *p* = 0.983; SOD: *p* = 0.070). IFN-α levels in PBMC cultures stimulated with R848 showed no notable change from D120 to D180 (*p* = 0.221). However, there was a significant increase in IL-6 production by PHA-L-stimulated PBMCs (*p* = 0.001), whereas IL-1β levels did not exhibit a significant difference (*p* = 0.387). TNF-α levels showed a significant increase from D120 to D180 (*p* = 0.003).

After observing some immunological changes in the patients in the syrup group, we analyzed the results from all the patients who were taking the ELA blend twice a day, in both the capsule and syrup forms (Appendix A). There was a statistically significant increase in CRP levels before versus after supplementation (1.1 mg/L vs. 1.4 mg/L, *p* = 0.045), with a median difference of 0.085 mg/L (CI: −0.01 to 0.33). However, this difference, while statistically significant, may not be clinically significant, due to the small magnitude of change. No significant changes in SOD activity, IFN-α, and IL-6 levels were seen before and after supplementation. Significantly, the levels of IL-1β production by PHA-L stimulated PBMCs increased from 12.1 pg/mL to 14.3 pg/mL following supplementation (*p* = 0.018). The median difference was 5.4 pg/mL (CI: −2.0 to 11.2). In the same way, TNF-α production levels increased significantly following supplementation, with the median going from 170.5 pg/mL to 196.3 pg/mL (*p* = 0.015). The median difference was 66 pg/mL (CI: 20.4 to 82.5). Overall, the study highlights the potential immunomodulatory effects of the ELA blend, particularly in enhancing specific aspects of the immune response characterized by increased cytokine production.

### 2.4. Influence of ELA Blend on Cytokine Production in Stimulated and Unstimulated PBMCs

Additional in vitro analysis revealed the effect of an ELA blend at varying concentrations (10 μg/mL and 150 μg/mL) on cytokine production by PBMCs, in comparison to a control group (untreated cells). For all the tested cytokines, both concentrations of ELA increased their production compared to the control group, with a more significant effect observed at 150 μg/mL (IFN-γ, *p* = 0.020; IL-1β, *p* = 0.004; IL-2, *p* < 0.001; IL-6, *p* < 0.001; TNF-α, *p* < 0.001; Table 5). Notably, TNF-α production showed the most substantial increase with the ELA blend at 150 μg/mL compared to the control, highlighting the potency of this blend.

The experimental study utilized two distinct stimulants, PHA-L (leading to T-cell activation and proliferation) and R848 (used to simulate viral infection and induce innate immunity), to activate PBMCs and investigate the immunomodulatory effects of the ELA blend at varying concentrations (10 μg/mL and 150 μg/mL) (Table 6). When PBMCs were treated with 10 μg/mL of ELA prior to stimulation with either PHA-L or R848, no significant differences were observed in the production of IFN-γ, IL-1β, IL-2, IL-6, or TNF-α compared to stimulated cells alone (*p* > 0.5).

Stimulation with PHA-L combined with ELA did not lead to detectable levels of IFN-α. However, pretreatment with 10 μg/mL ELA before R848 stimulation significantly increased IFN-α production compared to R848 stimulation alone (*p* = 0.012). Treatment with 150 μg/mL ELA did not significantly affect IFN-α production compared to R848 stimulation (*p* > 0.999). Additionally, 150 μg/mL ELA significantly reduced IFN-γ (*p* = 0.008) and IL-2 (*p* = 0.012) levels compared to PHA-L stimulation. However, this concentration of ELA did not significantly affect IFN-γ production compared to R848 stimulation (*p* = 0.267). Treatment with 150 μg/mL ELA induced significant IL-1β production compared to PHA-L stimulation (*p* = 0.005), but significantly reduced IL-1β levels compared to R848 stimulation alone (*p* < 0.001). Neither dose of ELA significantly impacted IL-6 production compared to PHA-L stimulation (*p* < 0.999 for both doses). However, ELA at 150 μg/mL significantly reduced IL-6 production compared to R848 stimulation alone (*p* < 0.001). Lastly, TNF-α production in response to PHA-L stimulation was not significantly affected by ELA treatment at either concentration. Conversely, ELA at 150 μg/mL significantly reduced TNF-α production compared to R848 stimulation (*p* = 0.049).

These results highlight the varying immunomodulatory effects of the ELA blend on cytokine production in PBMCs under different immune-stimulated conditions.

### 2.5. ELA Blend Improves Innate Antiviral Immune Response of Peripheral Blood Leukocytes (PBLs) Ex Vivo

The study investigated the impact of the ELA blend on innate antiviral immune response, using a test based on the resistance of freshly isolated PBLs ex vivo to indicatory vesicular stomatitis virus (VSV) infection and cytokine production. First, the cytotoxicity of the ELA blend was investigated. To determine the highest non-toxic concentration of the ELA blend for PBLs, cells were cultured for 24 h with varying concentrations of the ELA blend (0–600 µg/mL) at 37 °C with 5% CO_2_. Subsequently, Trypan Blue staining was performed to evaluate cell viability. The results (Figure 1A) indicated that 150 μg/mL of the ELA blend was the highest non-toxic dose, with cell viability reaching 80% compared to control cells. The cytotoxic concentration for 50% of living cells (CC50) was 594.6 µg/mL. Simultaneously, no virucidal effects against VSV were observed at this concentration of the ELA blend. The negative control was the virus incubated in an appropriate culture medium (VSV), and the positive control was the virus incubated with 50% ethanol (ALK50). The experiment was performed three times (n = 3). The criterion for determining the virucidal activity of the preparation was the ability to reduce the infectious titer of the virus (TCID50—tissue culture infectious dose) by 4 logs (i.e., 99.99%) (Appendix A).

In a series of 10 experiments, in three repetitions each, involving isolated PBLs ex vivo from random individuals, the findings revealed that the ELA blend at a concentration of 150 µg/mL (*p* = 0.0156 for binomial test) improved innate antiviral immune response by decreasing VSV replication. This observation strongly suggests that the ELA blend exerts a discernible influence on VSV replication, leading to significant reduction in viral titer (median of diff. −1.67 and CI –3.0 to -0.3, *p* = 0.031). The results are presented in Figure 1B.

Furthermore, we investigated the impact of the ELA blend on cytokine production by VSV-infected PBLs ex vivo. Supernatants from the infected PBL cultures were collected after 24 h of incubation and analyzed using both the Milliplex-magnetic bead method and ELISA. Our analysis revealed that the ELA blend administered at a concentration of 150 µg/mL elicited a significant increase in IFN-γ (66 pg/mL, IQR = 144, *p* = 0.002), IL-1β (276.7 pg/mL, IQR = 110.7, *p* = 0.004), and TNF-α (4399 pg/mL, IQR = 1260, *p* = 0.002) production by VSV-infected leukocytes. Conversely, in the case of IL-10, we observed that the ELA blend reduced IL-10 production (472.5 pg/mL, IQR = 695.2, *p* = 0.037) in VSV-infected cells. The results are presented in Figure 1C–F. We did not observe any changes in IL-6 concentration following the treatment with the ELA blend. Interestingly, the concentration of IFN-α was found to be below the detection limit.

## 3. Discussion

URTIs pose significant public health challenges due to their high prevalence and associated morbidity. In this study, we investigated the potential effects of an ELA blend, containing extracts of EP, LCK, and AM, on patients predisposed to URTIs. The results from our in vitro study indicate that the immunomodulatory effects of the ELA blend are dependent on dosage, duration of treatment, and the type of immune stimulus used. This variability underscores the complex nature of the interactions of natural compounds with immune cells, and highlights their potential for both anti-inflammatory and immunostimulatory effects, depending on the context. Our findings provide valuable insights into the immunomodulatory and symptom-relieving properties of this natural formulation.

One of the key findings of our study was the significant reduction in URTI symptoms among patients receiving the ELA blend compared to the placebo. Notably, fewer patients in the ELA blend group reported concurrent symptoms such as runny nose, cough, fever, and difficulty breathing. The prevention and alleviation of the symptoms of URTIs has been a subject of interest in the field of herbal medicine. *Echinacea purpurea*, in particular, has been the target of numerous studies evaluating its effectiveness in preventing and treating URTIs. Research has shown that *Echinacea* extracts may be beneficial for the early treatment of existing URTIs. Additionally, *Echinacea* has been reported to reduce the average length of URTI infections, both bacterial and viral, indicating its potential in alleviating symptoms [18]. Furthermore, *Echinacea* preparations are widely used and marketed for the treatment and prevention of common colds and URTIs [19]. In a clinical trial, participants taking a combination of Echinacea–Zinc–Vitamin C experienced a significant reduction in the severity of cold symptoms compared to those receiving a placebo [20]. Moreover, a meta-analysis concluded that *Echinacea* could decrease the duration of cold symptoms by approximately 1.4 days [21]. Despite the promising data, the efficacy of *Echinacea purpurea* is not universally accepted, and some studies have reported inconclusive results. For example, a study found no significant difference in the rate of infection or time to first infection among participants taking *Echinacea* compared to those on a placebo [22]. This inconsistency may be attributed to variations in *Echinacea* formulations, dosages, and study designs, which complicate the interpretation of clinical outcomes [23]. Furthermore, the possible health benefits of AM have been investigated, especially in relation to URTIs and colds. A 4-week trial with 11 women showed that dietary supplementation with *Aronia melanocarpa* extract (150 mg/day) significantly increased body surface temperature and improved psychological factors related to cold constitution, though it did not affect peripheral blood flow. These effects may be mediated by elevated plasma noradrenalin levels and reduced oxidative stress [24]. A randomized, controlled, double-blind, parallel intervention trial involving healthy adults demonstrated that daily consumption of 100 mL of *Aronia melanocarpa* juice for 30 days prevented increases in cholesterol levels, reduced postprandial glucose, and induced metabolomic shifts indicative of decreased inflammation [25]. A pilot study on nursing home residents found that black chokeberry juice reduced the incidence of urinary tract infections (UTIs) requiring antibiotics by 55% and 38% in two groups during a 3-month administration period, although no immediate effects or changes in other infections were observed [26]. Many studies highlight that honeysuckle berries possess exceptionally high antioxidant activities, primarily due to their rich content of bioactive compounds such as vitamin C, polyphenols, and anthocyanins, which effectively combat oxidative stress and offer significant protective benefits against related health issues [27]. Currently, there is a lack of clinical data demonstrating the efficacy of *Lonicera cerulea* in any treatment.

The immunostimulatory effects of polyphenols have been extensively studied, revealing their potential to modulate immune responses. Polyphenols have been shown to regulate immune function by promoting immunity to foreign pathogens, inducing innate immune responses, and rescuing immune response homeostasis [28,29]. Our study demonstrates that the ELA blend, particularly at a concentration of 150 μg/mL, exhibits significant immunostimulatory properties by inducing cytokine production in unstimulated PBMCs. In particular, ELA treatment prior to stimulation with toll-like receptor 7 and 8 agonist (R848) resulted in a notable increase in IFN-α and IL-2 production compared to cells stimulated with R848 alone. However, ELA treatment did not affect IFN-γ production in this context. Importantly, pretreatment with ELA before R848 stimulation effectively reduced the production of IL-1β, IL-6, and TNF-α compared to cells stimulated with R848 alone. When using PHA-L as a stimulus after ELA treatment, we observed a significant decrease in IFN-γ and IL-2 production, while IL-6 and TNF-α production remained unaffected. In comparison, unlike the ELA blend, another study showed that walnut polyphenolics had no effect on cytokine secretion from resting PBMCs. However, walnut polyphenolics and ellagic acid, known for their antioxidant and anti-inflammatory properties, modulated cytokine production in PHA-stimulated PBMCs. Specifically, walnut polyphenolics inhibited cellular proliferation while increasing IL-2 production and reducing IL-13 and TNF-α levels [30]. This aligns with Mlcek et al.’s findings, in which quercetin was shown to inhibit the release of pro-inflammatory cytokines (IL-6, IL-8, TNF-α) from mast cells stimulated with phorbol 12-myristate 13-acetate (PMA) and calcium ionophore. Moreover, quercetin reduced phorbol calcium ionophore-induced activation of NF-κB [31]. Furthermore, research has demonstrated that polyphenols’ effect depends on the type of immunological cells; e.g., green tea catechin metabolites can increase NK cell cytotoxicity [32], while quercetin has been found to enhance NK cell lytic activity in animal models [33]. Moreover, clinical trials have shown that dietary interventions involving polyphenol-rich juices can enhance NK cell function, including increased lymphocyte proliferative responsiveness, IL-2 secretion, and NK cell lytic activity in healthy participants [34]. Our study’s observations of increased IFN-α and IL-2 production in PBMCs following ELA treatment with R848 stimulation aligns with the broader understanding of polyphenols and their immunomodulatory effects. While our study specifically focused on PBMCs, future investigations could explore the impact of ELA blends on NK cell activity and cytotoxicity, potentially contributing to enhanced immune surveillance and anti-tumor responses.

Our investigation showed strong potential for the ELA blend to stimulate an innate antiviral immune response. It inhibited VSV replication in PBLs when administered after viral infection. VSV was selected as the indicator virus in the test of PBLs’ resistance to viral infection, given that it is an animal virus and does not naturally infect humans [35]. The observed beneficial effect of the ELA blend was accompanied by a marked increase in IFN-γ, IL-1β, and TNF-α production by PBLs. These investigated cytokines play crucial roles in innate antiviral immune response, highlighting the potential of the ELA blend to enhance innate immune defenses against viral infections. The observed elevation in IFN-γ production suggests that the ELA blend stimulates cellular immunity, contributing to the activation of antiviral pathways. IFN-γ is known for its role in coordinating immune responses against viral infections by promoting the activity of immune cells such as macrophages and natural killer cells [36]. Additionally, the increase in IL-1β and TNF-α levels signifies an immunostimulatory effect induced by the ELA blend. IL-1β and TNF-α are essential mediators of inflammation and host defense, orchestrating immune cell recruitment and activation to combat viral threats [37]. Conversely, the ELA blend demonstrated an antagonistic effect on IL-10, showing a reduction in production by VSV-infected PBLs. IL-10 is an anti-inflammatory cytokine that can suppress immune responses. The reduction in IL-10 production caused by the ELA blend suggests a shift toward a more pro-inflammatory immune environment conducive to antiviral defense. Comparing our findings with the immunomodulatory effects of almond skin extract, as described in previous research [38], underscores the role of polyphenols in mediating these responses. Arena et al. showed that Human herpesvirus type 2 (HHV-2) infection increases the production of IL-17 and IL-4 in PBMCs, while almond skin extract stimulates the production of IFN-γ, IFN-α, TNF-α, IL-12 and IL-4, with IL-4 being produced to a greater level than with HHV-2 alone [39]. Almond skin extract treatment significantly decreased the production of IL-17 induced by HHV-2 and increased the production of IFN-γ, IFN-α, TNF-α, IL-12 and IL-4 in PBMCs, regardless of HHV-2 infection status. However, it is noteworthy that the effects of almond skin extract on IL-10 production observed in previous studies appear to be contradictory to our result and its documented antiviral activity [17]. We previously demonstrated that the ELA blend effectively inhibited viral replication without a virucidal effect, achieving an 87.5% reduction in viral titer against HCoV-OC43, highlighting its potential as an antiviral agent [10]. This post-infection mode of action aligns with the results observed for a blend of double-standardized extracts of *Aronia melanocarpa* (Michx.) Elliot and *Sambucus nigra* L. (EAM-ESN), which also primarily exerts its antiviral effects after viral entry into host cells. The EAM-ESN blend, however, exhibited a broader spectrum of activity, strongly inhibiting both HCoV-OC43 and A/H1N1, while its effects on HHV-1 and HAdV-5 were limited [6]. Moreover, the *Echinacea purpurea* preparation exhibited virucidal activity primarily through direct contact with the virus. It effectively inactivated HCoV-229E, MERS-CoV, and SARS-CoV-1 at similar concentrations, showing a straightforward mechanism of viral inactivation without significant effects on cytokine production. The observed virucidal effect against SARS-CoV-2 at a concentration of 50 µg/mL suggests that the *Echinacea* preparation may be more suitable as a primary prophylactic treatment aimed at direct virus inactivation [40]. This discrepancy underscores the complexity of botanical extracts and their multifaceted effects on immune regulation. The ELA blend not only possesses direct virucidal properties, but also appears to modulate the immune response, as evidenced by the increased production of IFN-γ, IL-1β, and TNF-α in PBLs, which are key cytokines involved in antiviral defense. This suggests that the ELA blend could enhance the host’s innate immune response, potentially offering a broader therapeutic application beyond simple viral inactivation.

It is important to acknowledge the limitations of our study, including the relatively small sample size and the short duration of the observation period. Future research with larger cohorts and longer follow-up periods could provide more robust evidence of the clinical efficacy of the ELA blend in preventing and treating URTIs. Additionally, for patients who initially received placebo capsules followed by the ELA blend syrup, it is crucial to conduct a study using a randomized, placebo-controlled crossover design. This approach would help to minimize bias and better assess the specific effects of the ELA blend compared to a placebo. Another limitation is the lack of etiology for symptoms. Patients were not tested to differentiate the type of infection or pathogen causing the symptoms.

A significant strength of this research lies in its rigorous methodology. The study was conducted under the supervision of an independent Contract Research Organization (CRO), ensuring adherence to international standards of clinical research. To minimize potential biases, the trial employed a double-blind design. Neither the medical investigator nor the laboratory personnel analyzing the biological samples were aware of the group allocations of the participants. This approach ensured the objectivity of both clinical observations and laboratory assessments. By incorporating both clinical and in vitro analyses, this study offers comprehensive insights into the blend’s antiviral and immunomodulatory properties. For the in vitro component, we utilized PBMCs and PBLs isolated from healthy donors. These cell types closely mimic the physiological immune responses seen in vivo, providing a robust model to investigate the immunomodulatory effects of the ELA blend. The use of a market-available, standardized product not only facilitates a broader application of these findings, but also ensures that the results are directly translatable to real-world scenarios.

## 4. Materials and Methods

### 4.1. Human Study Protocol

The study (NCT06020001) enrolled a cohort of 61 participants, of whom 59 successfully concluded the research. The study was conducted at a single research center in Poland (Sopot). The first participant was enrolled on 17 October 2021, and the last participant was enrolled on 2 November 2022. The study was conducted in accordance with the protocol (0122-PMC-COV) and ethics committee opinion number KB-4/91/2022/29.10.2022, issued by the Regional Medical Chamber in Gdańsk on 29 September 2022, with approved modifications on 6 February 2023. The registration of the trial was delayed due to miscommunication between the CRO and the Sponsor. Once the issue was identified (following the study’s completion and the decision to pursue publication), the trial was promptly registered. This delay did not impact the study’s design, conduct, or data integrity, and full transparency has been maintained throughout.

Participants for the study were recruited under the supervision of the principal investigator, M.D. Dagmara Wiewiórkowska-Garczewska. Recruitment was conducted through structured medical interviews, which assessed the volunteers’ medical history and current health status. The participants were not required to stay at the research center continuously during the study period. Instead, regular visits were scheduled to monitor compliance, collect data, and perform symptom assessments.

The initial phase of the trial involved the allocation of patients into two distinct groups: the placebo group and the ELA blend group. The patients were allocated to a group using block randomization, with a block size that was randomly even. To implement the randomization list, the statistical software R (version 3.6.3) was utilized, employing the blockrand package (version 1.5). Block randomization was prepared by an investigator with no clinical involvement in the trial. The participants were administered the experimental composition or placebo in the form of capsules for a duration of 60 days. Subsequently, a 60-day observation period (follow-up) was conducted after the termination of supplementation. Blood specimens were obtained from individuals on day D0 (before the start of supplementation), D60 (after 60 days of supplementation), and D120 (after 120 days from the start of supplementation). Both the physician and patients were kept blind to the allocation. In the subsequent phase of the investigation, individuals who were administered a placebo were administered the experimental formulation in the form of syrup. Fresh informed consent was acquired, and the duration of supplementation remained consistent with that of the capsules (60 days). In the trial, the patients who were administered the second formulation were subjected to an additional blood sample collection on day 180 (D180). During the course of the trial, the participants completed patient diaries to document their symptoms and severity of sickness, along with any supplementary medications consumed and the reason for their use. Two participants in the capsule group experienced digestive issues, which resulted in their withdrawal from the study. One patient discontinued their participation in the second phase of the trial due to a surgical procedure.

The study included participants aged 18–70 years, both women and men, who were healthy at the time of inclusion, with no active upper respiratory tract inflammation. It also included individuals with chronic upper respiratory tract diseases, such as asthma, and those who had experienced two or more episodes of upper respiratory tract infections or two or more colds per year. The participants provided written informed consent, were able to maintain a paper patient diary, and attended all patient visits. The study excluded individuals who had taken supplements containing plant extracts, polyphenols, or anthocyanins (berry fruits) within 3 months before the study, those with acute inflammation, and those who regularly used oral immunosuppressive drugs. It also excluded individuals who had used steroids or anti-allergic medications for chronic disease exacerbations within at least 2 months before the study, those who had used or completed antibiotic therapy within 1 month before the study, and those participating in another clinical study. Additionally, the study excluded individuals with cancer, autoimmune disease, severe liver dysfunction, tuberculosis, leukemia, multiple sclerosis, AIDS, rheumatoid arthritis, or organ transplants, as well as those who were unable to swallow the oral study drug/placebo. Participants with hypersensitivity or allergy to any ingredient of the preparation, especially plants from the Asteraceae family (e.g., chamomile, calendula, yarrow), and pregnant women, or those planning to become pregnant during the study period, were also excluded.

### 4.2. Medications

The ELA blend was provided by GREENVIT Ltd. (Zambrów, Poland). Pure extracts were produced by water or ethanol extraction under mild conditions, to preserve thermolabile compounds. The ELA blend was standardized to contain a minimum of 15% anthocyanin content, as measured by HPLC; a minimum of 30% polyphenol content, as measured by UV; and a minimum of 0.2% chicoric acid content, as measured by HPLC. The preparation and chemical analysis of the ELA blend have been previously described [10]. The components were blended in precise ratios to meet the criteria for standardizing the final product. This proprietary formulation, developed by AronPharma Ltd. (Gdansk, Poland) is pending patent approval.

The placebo used in this study was formulated with maltodextrin, providing a comparable appearance to the ELA blend capsules. Both the ELA blend and the placebo were prepared in the form of hydroxypropyl methylcellulose (HPMC) capsules, with each package containing 60 capsules. They were pre-packed in bottles and consecutively numbered for each participant according to the randomization schedule. Each participant was assigned an order number and received the capsules in the corresponding pre-packed bottle. The storage temperature for both formulations was maintained between 18 and 24 °C (room temperature). Additionally, the ELA blend was also available in a syrup formulation (second phase of the study). It was administered at a dosage of 5 mL twice a day, with each study participant receiving two 150 mL bottles of the syrup. In both formulations, the dosage of the ELA blend was 300 mg twice a day. The ELA blend was administered either as capsules or as a syrup, with participants instructed to take one capsule or 5 mL of syrup in the morning and one capsule or 5 mL of syrup in the evening.

### 4.3. Patient Diaries

The participants completed patient diaries (Appendix A) during the observation period (D1-D120 for the capsule group and D1-D180 for the syrup group). They were instructed to record any symptoms or illnesses that manifested throughout the day in the journal. The intensity of symptoms was evaluated using a numerical scale ranging from 0 to 4, with 0—no symptoms, 1—mild symptoms, 2—moderate symptoms, 3—severe symptoms, and 4—extremely severe symptoms. The diary documented various symptoms associated with respiratory tract infections, including runny nose, nasal congestion, sneezing, fever, sore throat, hoarseness, throat scratching, cough, weakness, tiredness, muscle ache, headache, and breathing difficulties. Additionally, participants had to report other symptoms such as abdominal pain, vomiting, diarrhea, constipation, nausea, rash, and dry mouth. If they experienced symptoms other than those specified, participants were requested to provide a description of the symptom and indicate its intensity.

Symptoms were categorized into clinically relevant groups to reflect common presentations of upper respiratory conditions or potential treatment side effects. Participants were considered to have “reported” symptoms if they documented their occurrence in their symptom diaries. These diaries were designed to capture the daily presence and severity of symptoms throughout the study.

The symptom data were aggregated and analyzed over two observation periods: D1-D60 (during treatment) and D61–D120 (post-treatment). Participants were included in specific symptom analyses only if they had complete and non-missing entries for the relevant symptom groups. Two participants were excluded from the analysis due to substantial missing data, ensuring the reliability of the reported statistics.

The outcomes in Table 2 were calculated as the number of participants who reported at least one occurrence (one day) of the symptom or any symptom from symptom group during the specified period. Additionally, the incidence of symptoms was further analyzed based on the number of patients reporting symptoms multiplied by the number of days on which the symptoms were reported. For example, if a single participant reported fever on three separate days, this was recorded as three incidents of fever. This methodology quantifies both the number of participants experiencing a symptom and the duration and frequency of the symptom within the study period.

### 4.4. Patient Samples Analysis

For the serum parameter evaluation, blood serum samples were collected from patients to quantify C-reactive protein (CRP) levels and assess superoxide dismutase (SOD) activity. The concentration of CRP was determined using a high-sensitivity Human C-Reactive Protein/CRP DuoSet ELISA kit (R&D Systems, Inc., Minneapolis, MN, USA). SOD activity in the serum samples was analyzed using a Superoxide Dismutase Activity Assay Kit (#CS0009; Sigma-Aldrich, Saint Louis, MO, USA).

In parallel, peripheral blood mononuclear cells (PBMCs) were isolated from the patient blood samples using BD Vacutainer^®^ CPT™ tubes (Becton Dickinson, Franklin Lakes, NJ, USA). PBMCs were then seeded at a concentration of 1 × 10^6^ cells/mL. Some of the PBMCs were stimulated with 5 μg/mL phytohemagglutinin-L (PHA-L; Sigma-Aldrich, Saint Louis, MO, USA) for 5 h, while others were stimulated with 1 μg/mL resiquimod (R848; Sigma-Aldrich, Saint Louis, MO, USA) for 20 h, to induce immune responses. Following stimulation, supernatants from the PBMC cultures were collected and stored, frozen, for subsequent analysis. Cytokine levels in the supernatants were measured using Human Interleukin DuoSet^®^ ELISA Development Systems (R&D Systems, Inc., Minneapolis, MN, USA), following the manufacturer’s instructions. Absorbance readings were obtained at 450 nm and 570 nm (for background) using a Perkin Elmer EnVision 2103 Multilabel Reader (Perkin Elmer, Waltham, MA, USA).

### 4.5. In Vitro Study

#### 4.5.1. Isolation and Culture of Human Peripheral Blood Mononuclear Cells (PBMCs)

Human PBMCs were isolated from buffy coats obtained from healthy blood donors (from the Regional Center for Blood Donation and Blood Treatment in Gdańsk, Poland) (consent number: Dyr.M/073/03/AJC/2021) using a standard Ficoll (1.077 g/L) (Biowest, Nuaillé, France) density gradient centrifugation method. 2 × 10^6^ cells/mL were cultured in Roswell Park Memorial Institute (RPMI) 1640 medium (Biowest, Nuaillé, France), supplemented with 10% Fetal Bovine Serum (FBS; Biowest, Nuaillé, France), 100 U/mL penicillin, and 100 µg/mL streptomycin (Sigma-Aldrich, Saint Louis, MO, USA). To assess the impact of the ELA blend on PBMC cytokine production, PBMCs were incubated with concentrations of 10 μg/mL and 150 μg/mL of the ELA blend for 20 h. Additionally, to evaluate the effects of immune stimulants (R848 and PHA-L) on PBMC activity, cells were treated with 1 μg/mL of R848 for 20 h or 5 μg/mL of PHA-L for 5 h. To investigate the potential modulatory effects of the ELA blend on R848- or PHA-L-induced PBMC responses, cells were pretreated with concentrations of 10 μg/mL and 150 μg/mL of the ELA blend for 4 h before incubation with R848 (20 h) or PHA-L (5 h). Following incubation, the culture medium with the PBMCs was collected, and cytokine levels were measured using Human DuoSet^®^ ELISA Development Systems (R&D Systems, Inc., Minneapolis, MN, USA), according to the manufacturer’s instructions.

#### 4.5.2. Isolation of Peripheral Blood Leukocytes (PBLs)

Peripheral blood samples were obtained from healthy volunteers at the blood donation center in Wrocław (Military Blood Donation and Blood Treatment Center, Wrocław, Poland). The blood samples were collected into tubes containing heparin (10 U/mL) to avoid clotting. Within an hour of collection, 5 mL of blood was layered onto 3 mL of Gradisol G (Aqua-Med, Poznań, Poland), then centrifuged at 400× *g* for 30 min. The leukocyte layer at the interface was collected, washed three times with complete medium RPMI 1640 (HIET PAS, Wroclaw, Poland) supplemented with 2% FBS (Sigma-Aldrich, Saint Louis, MO, USA), glutamine, and antibiotics (Sigma-Aldrich, Saint Louis, MO, USA), and resuspended in complete medium to a final concentration of 1 × 10^6^ cells/mL.

#### 4.5.3. Viability Assessment

Isolated PBLs (n = 3) were treated with a filtered (0.22 µm pore size; Merck Millipore, Burlington, MA, USA) test solution of the ELA blend at final concentrations of 600 µg/mL, 150 µg/mL, 37.5 µg/mL, and 0 (control). The cells were then incubated at 37 °C and 5% CO_2_ for 24 h. After incubation, 10 µL of cell suspension was mixed with 10 µL of 0.4% Trypan Blue solution (BioRad, Hercules, CA, USA). The stained cells were then loaded onto a two-chamber slide, which was placed in an automatic cell counter TC20 (BioRad, Hercules, CA, USA). Leukocyte viability was measured, taking into account cells in the range of 5–16 μm.

#### 4.5.4. Vesicular Stomatitis Virus (ATTC VR-1238™, Rhabdoviridae)

A wild-type Indiana VSV (vesicular stomatitis virus, Rhabdoviridae) serotype was used. The VSV was obtained from Dr. C. Buckler (National Institutes of Health, Bethesda, MD, USA). The virus was grown and titrated in L929 cells. The viral titer was expressed with reference to the TCID50 (tissue culture infectious dose) value, based on the cytopathic effect caused by this virus in approximately 50% of infected cells. L929 (ATCC CCL1), a murine fibroblast-like cell line, was maintained in complete RPMI 1640 medium (HIIET, Wroclaw, Poland) with antibiotics (100 U/mL penicillin and 100 μg/mL streptomycin), 2 mM L-glutamine, and 2% FBS (all from Sigma-Aldrich, Saint Louis, MO, USA).

#### 4.5.5. Assessment of PBLs’ Resistance to VSV Infection

PBLs (1 × 10^6^ cells/mL) were suspended in complete medium (RPMI 1640 medium with 2% FBS) and infected with VSV at a dose of 100 TCID50/mL. After 40 min of adsorption at room temperature, the virus was washed out three times with RPMI 1640 medium containing 2% FBS, and the cells were then suspended in 1 mL of complete medium. A sample of infected cells was kept at 4 °C as a control to establish the initial virus level. The remaining cells were incubated at 37 °C, 5% CO_2_. Subsequently, the virus titer and the levels of specific cytokines were measured in the supernatants of the PBLs.

#### 4.5.6. Cytokine Level Determination

Levels of the selected cytokines interleukin (IL)-10, interferon gamma (IFN-γ), IFN-α, IL-1β, and TNF-α were measured using the Milliplex^®^-magnetic bead system (Merck Millipore, Burlington, MA, USA). Supernatants collected from the infected PBL cultures were thawed and immediately applied to a Human Cytokine Panel A plate (Merck Millipore, Burlington, MA, USA), according to the manufacturer’s instructions. After overnight incubation at 8 °C, the plate was processed, and mean fluorescence intensity (MFI) was measured using a Luminex 200™ System (Merck Millipore, Burlington, MA, USA). Additionally, cytokine levels were measured using Human IL-1β, IL-2, and IL-6 DuoSet^®^ ELISA Development Systems (R&D Systems, Inc., Minneapolis, MN, USA), according to the manufacturer’s instructions.

### 4.6. Statistical Analysis

Statistical analysis was performed using GraphPad Prism 8 software (San Diego, CA, USA) and statistical software R (version 3.6.3). Quantitative variables that did not follow a normal distribution based on the Shapiro–Wilk test were presented using the Median and Interquartile Range (IQR). For comparing variables between different groups (e.g., ELA blend group vs. placebo group), unpaired statistical tests were used (Mann–Whitney U test or Fisher’s exact test). When comparing variables within the same group before and after treatment (e.g., baseline vs. post-treatment), paired statistical tests were employed (Wilcoxon signed-rank test). The Friedman test was utilized to assess whether there were significant differences in cytokine production across multiple concentrations of the ELA blend under various stimulation conditions. Statistical significance was determined using a threshold *p*-value of less than 0.05.

## 5. Conclusions

Our study highlights the promising therapeutic effects of the ELA blend for both preventing and reducing the intensity of symptoms associated with URTIs. However, in the future, it is necessary to conduct broader studies to assess the efficacy and safety of the ELA blend. Specifically, conducting research on a larger population to investigate its preventive effect on URTIs, as well as its symptom-relieving effects, will be crucial. To optimize therapeutic outcomes, future investigations should focus on examining the blend’s impact on specific viral and bacterial pathogens associated with URTIs. Additionally, ongoing surveillance of adverse effects and interactions with conventional therapies is essential to establish the safety profile of natural product interventions for respiratory infections. In vitro findings underscore the specific antiviral and immunomodulatory effects of the ELA blend, which can vary depending on the type of immune stimulus used. Further investigations are necessary to elucidate the underlying mechanisms and optimize the therapeutic potential of the ELA blend in immune-related contexts.

## Figures and Tables

**Figure 1 ijms-25-13436-f001:**
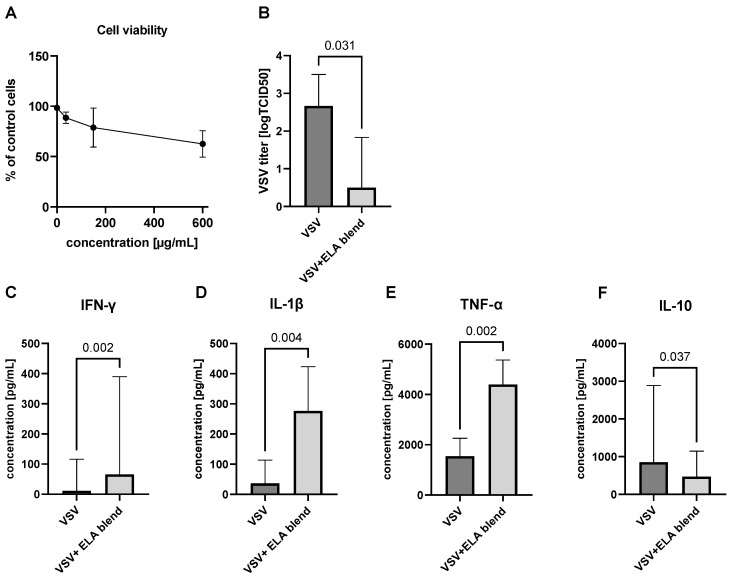
Impact of the ELA blend on vesicular stomatitis virus (VSV) replication/level of innate immunity after ELA blend treatment. (**A**) Cytotoxicity of the ELA blend on peripheral blood lymphocytes (PBLs). (**B**) The effect of the ELA blend was measured as the difference between the VSV titer (log TCID50/mL) after ELA treatment and the VSV titer (log TCID50/mL) before ELA treatment. (**C**–**F**) Impact of the ELA blend on cytokine production by VSV-infected PBLs ex vivo.

**Table 1 ijms-25-13436-t001:** Baseline characteristics.

Parameter		Placebo/Syrup	Capsules	*p* ^1^
Patients, n		30	31	
Sex, n	Women	19	21	0.791
	Men	11	10	
Age, median (IQR)		59 (19)	59 (12)	0.125
	Women	48 (21)	60 (12.5)	**0.005**
	Men	63 (12)	55 (24)	0.355
Reported colds/URTIs per patient during last 3 months before study, median (IQR)		5 (3)	4 (3)	0.585
Comorbidities, n	Any comorbidity	22	22	>0.999
	Hypertension	8	9	>0.999
	Diabetes	2	2	>0.999
	Allergic diseases	2	1	>0.999
Concomitant medications, n		22	22	>0.999

^1^ *p* values for Fisher’s exact test or Mann–Whitney test.

**Table 2 ijms-25-13436-t002:** Incidence of symptoms for patients who received the ELA blend in capsule form vs. the placebo during the observation period.

	Number of Patients		*p* ^1^
	Capsules	Placebo	
reporting at least three out of four symptoms at the same time (runny nose, cough, fever, difficulty breathing) during the observation period (D1–D120)	2	8	**0.044**
fever during treatment (D1–D60)	1	4	0.187
fever after treatment (D61–D120)	1	5	0.104
sore throat/hoarseness/scratching in the throat during treatment (D1–D60)	8	16	**0.038**
sore throat/hoarseness/scratching in the throat after treatment (D61–D120)	7	8	0.337
cough during treatment (D1–D60)	3	8	0.099
cough after treatment (D61–D120)	3	9	0.058
runny nose/nasal congestion/sneezing during treatment (D1–D60)	8	13	0.159
runny nose/nasal congestion/sneezing after treatment (D61–D120)	9	9	0.576

^1^ *p* values for Fisher’s exact test (n_CAPSULES_ = 28; n_PLACEBO_ = 29). Two patients were removed from the analysis due to a substantial amount of missing data in their symptom diaries.

**Table 3 ijms-25-13436-t003:** The impact of the ELA blend on immune system indicators compared to the placebo group. The study period spans from day D0 to D60. The dataset includes measurements of serum CRP concentration (C-Reactive Protein), SOD activity (Superoxide Dismutase), and the production of cytokines such as IFN-α (Interferon-alpha), IL-6 (Interleukin-6), IL-1β (Interleukin-1 beta), and TNF-α (Tumor Necrosis Factor-alpha) in PBMCs (Peripheral Blood Mononuclear Cells) stimulated with PHA-L (Phytohemagglutinin-L) or R848 (Resiquimod). Statistical evaluations are represented through Median (Me), Median of Differences (MEDIAN OF DIFF.), Interquartile Range (IQR), Confidence Interval (CI), and *p*-values (*p*).

Sample	Parameter	Group	D0			D60			D0 vs. D60		
			Me	IQR	*p* ^1^	Me	IQR	*p* ^1^	*p* ^2^	Median of Diff.	CI
serum	CRP [mg/L]	CAPSULES	1.2	2.4	0.25	1.7	9.1	0.17	**0.010**	0.43	0.08 to 1.03
		PLACEBO	0.7	1.7		0.9	2.1		0.080	0.25	0.00 to 0.74
	SOD [units/mL]	CAPSULES	39.5	10.8	0.824	43.3	11.5	0.303	0.219	4.7	−5.5 to 9.5
		PLACEBO	40.9	13.7		42.5	9.5		0.711	−1.7	−6.7 to 6.2
supernatant from PBMCs stimulated with R848	IFN-α [pg/mL]	CAPSULES	69.8	91.8	0.384	110.0	104.5	0.407	0.070	17.5	−3.6 to 61.2
		PLACEBO	94.6	83.7		111.8	144.0		0.070	33.8	−9.3 to 88.1
supernatant from PBMCs stimulated with PHA-L	IL-6 [pg/mL]	CAPSULES	570.5	852.4	0.738	484.4	655.8	0.501	0.256	−74.7	−421.0 to 185.3
		PLACEBO	498.0	364.2		403.7	539.9		0.584	−102.6	−146.4 to 127.8
	IL-1β [pg/mL]	CAPSULES	10.5	15.8	0.199	20.3	36.9	0.755	**0.027**	7.6	0.5 to 26.8
		PLACEBO	12.1	6.1		16.9	35.7		0.231	0.9	−3.5 to 31.6
	TNF-α [pg/mL]	CAPSULES	190.9	250.2	0.901	203.2	289.1	0.889	0.468	43.6	−55.1 to 86.7
		PLACEBO	183.0	306.7		219.8	168.5		0.612	10.55	−122.2 to 79.5

^1^ *p* value for a Mann–Whitney test, ^2^
*p* value for Wilcoxon test.

**Table 4 ijms-25-13436-t004:** The impact of the ELA blend on immune system indicators in patients receiving the ELA blend in the form of syrup. The dataset includes measurements of serum CRP concentration (C-Reactive Protein), SOD activity (Superoxide Dismutase), and the production of cytokines such as IFN-α (Interferon-alpha), IL (Interleukin)-6, IL-1β, and TNF-α (Tumor Necrosis Factor-alpha) in PBMCs (Peripheral Blood Mononuclear Cells) stimulated with PHA-L (Phytohemagglutinin-L) or R848 (Resiquimod). Statistical evaluations are represented through Median (Me), Median of Differences (MEDIAN OF DIFF.), Interquartile Range (IQR), Confidence Interval (CI), and *p*-values (*p*).

Sample	Parameter	Group	D120		D180		D120 vs. D180		
			Me	IQR	Me	IQR	*p* ^1^	Median of Diff.	CI
serum	CRP [mg/L]	SYRUP	0.9	2.0	0.6	1.7	0.983	0.02	−0.22 to 0.15
	SOD [units/mL]	SYRUP	100.8	13.0	99.3	8.9	0.070	−2.6	−7.9 to 1.9
supernatant from PBMCs stimulated with R848	IFN-α [pg/mL]	SYRUP	168.0	168.5	117.4	145.2	0.221	−9.4	−40.5 to 20.9
supernatant from PBMCs stimulated with PHA-L	IL-6 [pg/mL]	SYRUP	335.9	323.1	534.7	319.0	**0.001**	161.1	36.6 to 295.3
	IL-1β [pg/mL]	SYRUP	13.7	11.4	11.4	10.5	0.387	0.05	−5.2 to 7.6
	TNF-α [pg/mL]	SYRUP	115.1	152.7	194.1	108.9	**0.003**	72	28.9 to 98

^1^ *p* value for Wilcoxon test.

**Table 5 ijms-25-13436-t005:** The effect of the ELA blend at different concentrations (10 μg/mL and 150 μg/mL) on cytokine production by PBMCs (Peripheral Blood Mononuclear Cells) in vitro vs. in the control group (untreated cells). Statistical evaluations are represented through Median (Me), Interquartile Range (IQR), and *p*-values (*p*). Measurements below the detection limit are indicated as ‘bdl’.

	IFN-γ [pg/mL]			IL-1β [pg/mL]			IL-2 [pg/mL]			IL-6 [pg/mL]			TNF-α [pg/mL]		
	Me	IQR	*p* ^1^	Me	IQR	*p* ^1^	Me	IQR	*p* ^1^	Me	IQR	*p* ^1^	Me	IQR	*p* ^1^
control	bdl			bdl			bdl			bdl			35.6	20.6	
ELA [10 μg/mL]	125.5	40.4	>0.999	20.9	24.0	>0.999	36.6	84.5	0.267	45.2	38.7	0.267	81.5	20.6	0.267
ELA [150 μg/mL]	1500.0	1438.0	**0.020**	122.3	98.7	**0.004**	144.3	127.9	**<0.001**	239.4	320.6	**<0.001**	3209.0	1219.0	**<0.001**

^1^ *p* value for Friedman tests with post hoc test.

**Table 6 ijms-25-13436-t006:** The impact of the ELA blend pretreatment at varying concentrations (10 μg/mL and 150 μg/mL) on the production of cytokines by PHA-L (Phytohemagglutinin-L)- and R848 (Resiquimod)-stimulated PBMCs (Peripheral Blood Mononuclear Cells) in vitro. Statistical evaluations are represented through Median (Me), Interquartile Range (IQR), and *p*-values (*p*). Measurements below the detection limit are indicated as ‘bdl’.

		IFN-α [pg/mL]			IFN-γ [pg/mL]			IL-1β [pg/mL]			IL-2 [pg/mL]			IL-6 [pg/mL]			TNF-α [pg/mL]		
		Me	IQR	*p* ^1^	Me	IQR	*p* ^1^	Me	IQR	*p* ^1^	Me	IQR	*p* ^1^	Me	IQR	*p* ^1^	Me	IQR	*p* ^1^
0	PHA-L [5 μg/mL]	bdl			304.6	196.2		bdl			540.0	567.8		458.4	309.7		1272.0	633	
ELA [10 μg/mL]		bdl			257.1	189.0	0.423	bdl		>0.999	168.2	237.7	0.635	295.3	277.6	>0.999	683.4	741.9	0.160
ELA [150 μg/mL]		bdl			134.7	102.0	**0.008**	3221.0	810.0	**0.005**	48.5	59.7	**0.012**	374.3	235.5	>0.999	1751.0	1971.0	0.635
0	R848 [1 μg/mL]	17.95	76.3		3724.0	2352.0		1042	343.5		45.2	130.2		3941.0	791.0		7364.0	2456.0	
ELA [10 μg/mL]		89.8	214.8	** *0.012* **	3814.0	2696.0	0.907	835.2	416.7	0.635	32.6	54.8	>0.999	2982.0	782.0	0.160	6019.0	2566.0	0.907
ELA [150 μg/mL]		40.35	63.3	*>0.999*	5032.0	3160.0	0.267	203.5	129.5	**<0.001**	118.8	93.2	**0.025**	525.5	594.8	**<0.001**	4979.0	951.0	**0.049**

^1^ *p* value for Friedman tests with post hoc test.

## Data Availability

The data that support the findings of this study are available from the corresponding author upon reasonable request.

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
