# Peer review of "Therapeutic Potential of a Natural Blend of *Aronia melancarpa*, *Lonicera caerulea*, and *Echinacea purpurea* Extracts in Treating Upper Respiratory Tract Infections: Preliminary Clinical and In Vitro Immunomodulatory Insights"

_ijms, 2024, doi:10.3390/ijms252413436_

Round 1
Reviewer 1 Report
Comments and Suggestions for Authors
In this study, the authors evaluated the efficacy of ELA blends (derived from three distinct plant extracts, namely Aronia melanocarpa, Lonicera cerulea, and Echinacea purpurea) in alleviating the symptoms of upper respiratory tract infections. The research groups were constituted of volunteers, one of whom received a placebo. It is regrettable that the research methodology is flawed in a number of ways, is presented in an unclear manner, and requires further clarification. The experimental setup itself also gives rise to a number of uncertainties, including the reliance on the subjective feelings of the volunteers as the basis for the results.
Specific comments are provided below:
Line 17 - Please define what ELA is.
Line 52-57 - This section is more suitable for a discussion than an introduction.
Line 58 – the research hypothesis is absent.
Line 94 - Please clarify how the symptoms of sore throat, hoarseness, and throat scratching were differentiated. Please clarify which scientific scale was used to assess this symptom. It is important to note that patients' perceptions of discomfort may vary considerably. For instance, one individual may perceive the sensation as pain, while another may describe it as a scratchy throat.
Line 392 – Please provide details regarding the recruitment of volunteers, their stay during the study period (60 days), the number of individuals involved in symptom assessment, and the methodology used to calculate the results.
Line 392 - A period of 60 days is a considerable length of time. Please provide justification for the necessity of this supplementation period. During 60 days it is possible to develop even several upper respiratory infections episodes.
Line 449 - Symptoms and feelings are subjective and cannot be the basis for making a scientific judgement.
Reviewer 2 Report
Comments and Suggestions for Authors
After some revisions, the study conducted by Zima et al. can be considered for publication in IJMS.
Line 17: Please, provide the meaning of “ELA”.
Scientific names like “Aronia melanocarpa“, “Lonicera cerúlea”, and “Echinacea purpurea” should be in italics. Please, revise the whole manuscript.
You need to highlight your main results (values and statistical significance) in the abstract and provide some directions for further investigations.
The Introduction needs some improvements. Explore the international context of your work and its novelty and relevance. A stronger background should be given. More information about Aronia melancarpa, Lonicera caerulea, and Echinacea purpurea should be given.
The Results section is well presented. However, Discussion can be improved. Besides the limitation, you should highlight your study’s strengths (both in a separate section/subsection). The Discussion should be divided into subsections aligned with what you presented in the Results and further/deeper analysis should be given by citing more literature to compare with the obtained results. Discuss the availability of the study plant extracts internationally or where it is difficult to access them, what similar plant extracts can be accessed? What differentiates these plant extracts from those you studied?
The Materials and Methods section is fine. However, it would be best if you justified the representativeness of your sample size.
Conclusions are also adequate.
More citations should be included in the Reference list. As mentioned before, the Introduction and Discussion sections need to be improved substantially.
Round 2
Reviewer 1 Report
Comments and Suggestions for Authors
The authors have provided a comprehensive response to the concerns that I initially raised.
Reviewer 2 Report
Comments and Suggestions for Authors
Thank you for addressing all my comments.